
# Tsunami risk perception in Central and Southern Italy.
Lorenzo Cugliari[1], Massimo Crescimbene[1], Federica La Longa[1], Andrea Cerase [1,2],
Alessandro Amato[1], Loredana Cerbara[3]
[1]National Institute of Geophysics and Volcanology, 00153, Roma, Italy
[2] Department of Communication and Social Research, La Sapienza University, 00198, Roma Italy
[3]Institute for Research on Population and Social Policies, National Research Council, 00185, Roma, Italy
*Correspondence to*: Lorenzo Cugliari (lorenzo.cugliari@ingv.it)





**Abstract**
The Tsunami Warning Center of the National Institute of Geophysics and Volcanology (CAT-
INGV) has been promoting, since 2018, the study of tsunami risk perception in Italy. Between
2018 and 2021 the semi-structured questionnaire on the perception of tsunami risk was
administered to a sample of 5,842 citizens residing in 450 Italian coastal municipalities,
representative of more than 12 million people. The survey was conducted with the Computer
Assisted Telephone Interview (CATI) methodology, described in Cerase et al. (2019) who
published the results of the first pilot survey (about 1,000 interviews). The large sample and
the socio-demographic stratification give an excellent representation of the resident population
in the surveyed Italian coastal municipalities. Moreover, in 2021 an optimized version of the
questionnaire was also administered via Telepanel (a tool for collecting proportional and
representative opinions of citizens) representative of the Italian population, which included
1,500 people distributed throughout the country.
In this work we present the main results of the three survey phases, with a comparison among
the eight surveyed regions, and between the coastal regions and some coastal metropolitan
cities involved in the investigations (Rome, Naples, Bari, Reggio Calabria and Catania).
Data analysis reveals heterogeneous and generally low tsunami risk perception. Some seaside,
in fact, show a good perception of tsunami risk while others, such as in Apulia and Molise,
reveal a lower perception, most likely due to the long time elapsed since the last event and the
lack of memory. We do not find relevant differences related to the socio-demographic
characteristics (age, gender) of the sample, whereas the education degree appears to affect
people's perception. The survey shows that the respondents' predominant source of
information on tsunamis is the television and other media sources (such us newspapers, books,
films, internet etc.) while the official sources (e.g. civil protection, local authorities, universities
and research institutes) do not contribute significantly. Also, we found an interesting difference
in people's understanding of the words tsunami and maremoto, the local term commonly used
in Italy until the 2004 Sumatra event, which should be taken into account in scientific and risk
communication. The nationwide sample shows lower tsunami risk perception compared to the
average of the coastal sample, confirming the need for thorough information campaigns
directed to tourists.
Our results are being used to drive our communication strategy aimed at reducing tsunami risk
in Italy, to activate dissemination and educational programs (data driven), to fill the data gap
on tsunami risk perception in the NEAMTWS area, and to implement multilevel Civil
Protection actions (national and local, top-down and bottom-up). Not least, outputs can address
a better development of the Tsunami Ready program in Italy.



## 1. Introduction

The Mediterranean region is highly exposed to tsunami risk, as witnessed by several recent events (Yalçıner et al., 2017; Triantafyllou et al., 2021; Dogan et al. 2021), basin-wide or local historical events (Maramai et al., 2014; Papadopoulos et al., 2014; Solov′ev et al., 2000; Maramai et al., 2021), and by the recent assessment of seismically induced tsunami hazard (Sørensen et al., 2012; Basili et al., 2021). However, the tsunami risk in the Mediterranean is thought to be underrated, due to the low frequency of events, as for other regions of the world (Dawson et al., 2004; Dogulu et al., 2014; UNESCO-IOC, 2017; Amato, 2020; Necmioğlu et al., 2021). It is therefore important to raise awareness among people, as well as of local authorities responsible for civil protection measures and emergency management, and national / regional authorities.

The Sendai Framework for Disaster Risk Reduction 2015-2030 gives high attention to Early Warning Systems (EWS), suggesting to "invest in, develop, maintain and strengthen people-centered multi-hazard, multisectoral forecasting and early warning systems, …; develop such systems through a participatory process; tailor them to the needs of users, including social and cultural requirements, in particular gender…". The strong emphasis on people means that any communication strategy should be based on a preliminary assessment of people's knowledge, awareness and perception of the risk. Moreover, the "needs of users" must be studied and understood, to reach people and communities in the right way using the best language and communication channels, and to have an optimal response in case of an impending inundation.

The Italian Tsunami Alert Center (Centro Allerta Tsunami, CAT) of the Italian National Institute of Geophysics and Volcanology (INGV) is part of the national Tsunami Warning System called SiAM (Italian national warning system for tsunami of seismic origin), established in 2017 with a Prime Minister Directive (G.U. n.128 del 05-06-2017). The SiAM is coordinated by the Civil Protection national Department (DPC), and besides the CAT-INGV, includes ISPRA. This latter acts as Tsunami Service Provider (TSP) for UNESCO Member States of the NEAM region (Amato et al., 2021), as well as National Tsunami Warning Center (NTWC) and Tsunami Focal Point (TFP) for Italy. Among the tasks of CAT, besides the tsunami surveillance / warning and the hazard assessment, there are also scientific and risk communications activities on tsunamis. CAT manages a dedicated web site (www.ingv.it/cat/en/), where people can find information on tsunami hazard/risk, on the warning system, on historical events, news on projects, papers, campaigns and drills.
In this perspective, the activities of the CAT include the assessment of tsunami risk people's perception. These researches aim to improve risk analysis and decision making, develop methods for eliciting opinions about risk, provide a basis for understanding and anticipating possible public reactions to tsunami hazards, enhancing risk communication among lay people, technical experts and policy makers (Slovic et al., 1982; Slovic, 1987; Wildavsky and Dake, 1990; Slovic, 2001; Rippl, 2002).
This paper presents the data on tsunami risk perception collected in Central and Southern Italy between 2018 and 2021. The study involved the administration of a semi-structured questionnaire to a sample of 5,842 people in 450 coastal municipalities.




## 2. Studies on tsunami risk perception, a brief overview

Risk perception studies taking into account the socio-cultural and psychological aspect, assess people's response to natural hazards and the behaviors they would adopt in response to the risks. People's perceptions - individually or collectively - of a natural hazard are also influenced by individual factors such as personality, age, beliefs, gender, education level, knowledge, and culture (Slovic, 1982; Slovic and Peters , 2006; McIvor et al., 2009; McNeill et al., 2013; Wachinger et al., 2013).

Human behavior is driven by perceptions (Slovic, 1987) rather than scientific knowledge about "facts" (Renn, 1990). Therefore, it becomes strategic for those involved in risk mitigation and communication, to have in-depth studies on the process that influences our ability to assess the risk of a natural phenomenon (Slovic, 1982) like tsunamis. Tsunamis are known to be a phenomenon with a low probability of occurrence but high impact, able to produce devastating consequences that would affect large areas and have serious consequences for human lives (Behrens et al., 2021; Rafliana et al., 2022). The low occurrence frequency is one of the variables that directly influence the risk perception, which makes these phenomena, in some contexts such as the Mediterranean, out of the collective consciousness, often inducing authorities not to undertake effective actions to reduce the risk.

However, tsunamis' low frequency of occurrence does not reduce their destructive potential. Moreover, how important it is to study people's perceptions of natural hazards (Lindell, 2000; Paton, 2010; Wachinger, 2013; Bonaiuto, 2016), particularly tsunami risk perceptions, emerges in various studies conducted in at-risk countries that were affected by tsunamis, such as for example, the 2004 Indian Ocean tsunami or the 2011 Japan tsunami (Kurita et al., 2007; Sugimoto et al. 2010; Alam, 2016; Arias, et al., 2017; Akbar et al., 2020).

The historical catalog of tsunami effects in the Mediterranean (Maramai et al., 2014) cites over 200 documented events for the whole area. More recently, in the Euro-Mediterranean area, 26 earthquakes above magnitude 5.5 occurred at sea or near the coast between 2017 and July 2022, triggering the activation of CAT. Among them, 10 earthquakes generated an alert level for possible sea level change including 6 Advisory (possible sea level change, less than 1 meter) and 4 Watch (possible sea level change with estimated run-up values above 1 meter), two of which caused damages in Greece and Turkey (Yalçıner et al., 2017; Dogan et al., 2019; Cirella et al., 2020; Dogan et al., 2020; Triantafyllou et al., 2020)

In addition, variables to be considered from a comprehensive perspective include the large growth in population living along the Euro-Mediterranean coasts. This phenomenon, occurring since the Second World War and intensified in recent decades, also includes the development of tourist facilities and large industrial complexes. These intensive forms of settlements require multi-risk analytical hazard approaches where the socio-cultural and psycho-social aspect becomes prominent. In this framework, strengthening tsunami risk perception studies that survey the opinions of employees, daily commuters as well as seasonal workers and tourists who increase the coastal human presence and consequently the risk is needed. The need to assess tsunami risk perception has been highlighted by several authors as a key to improving emergency behaviors and minimizing population risk by limiting casualties and infrastructure damage (Ho et al., 2008; Martin et al., 2009; Ritchie and Roser, 2014; EMDAT, 2019).

## 3. The CAT-INGV tsunami risk perception studies

Amid increasing international attention to tsunami risk, CAT-INGV since 2018 has been promoting tsunami risk perception studies to provide oriented support to civil protection activities and develop data-driven, context-appropriate risk communication strategies.


Moreover, the CATI questionnaire's structuring and administration methodology make it an
excellent tool to collect a large, standardized, retraceable and cost-effective amount of data.
(Dawson et al., 2004, Cugliari et al., forthcoming). Furthermore, The questionnaire as a survey
method to study tsunami risk perception is widely used in the international context (see for
example Apatu, 2013; Sun, 2013; Lindell, 2015; Lindell, 2016; Jon., 2016; Fraser, 2016; Wei,
2017; Buylova, 2020)
**3.1 The tsunami risk perception questionnaire**

In this study we have used the questionnaire designed and described in Cerase et al., 2019 (and
available in the English version in the 2019 paper's SOM), consisting of 6 sections and 27
items that allow us to detect respondents' opinions regarding knowledge, tsunami risk
perception, tsunami representation, cultural attitudes toward risks (Douglas and Wildavsky,
1982), and through which channels respondents have been informed about tsunamis and would
like to receive an alert in case of tsunami.
The same questionnaire, implemented and administered in 2018 was subsequently
administered in 2020 and 2021 by the same CATI methodology to extend the coverage to six
more Italian regions and achieve better statistical representativeness.
**3.2 - Study area and sample characteristics**

The coastal belts are the most densely populated territory and where the largest urban centers
are developed. 20% of the EU's approximately 500 million inhabitants live in the coastal area,
and in 2018, in Italy, 28% of the total population (more than 17 million inhabitants) resided in
coastal municipalities. Between 1951 and 2011, the increase in coastal population was about
29%. Sardinia, Sicily, Apulia and Calabria together represent about 64% of the national
coastline. The Italian region with the most coastal population is Latium (due to the presence of
the municipality of Rome, a metropolitan city) followed by Sicily and Campania. Coastal areas
are also among the most densely populated with an average of 398 inhabitants per $km^2$,
compared to 167 in noncoastal areas. This is also due to the presence of large urban centers
including 10 regional capitals (Report ISTAT, 2020).

Our survey was carried out in three different phases. The first survey phase, concluded in 2018
(April 4 to May 4, 2018) covered Apulia and Calabria regions where 1021 questionnaires were
collected (Cerase et al., 2019). In the second survey phase, carried out between Dec. 27, 2019
and Jan. 8, 2020, 614 questionnaires were collected in the coastal municipalities of Molise,
Basilicata and Eastern Sicily. In the third survey phase, completed in 2021 (between Dec. 21,
2020 and Jan. 8, 2021), 4,207 questionnaires were collected in the coastal municipalities of
Latium, Campania, Sardinia, southern and northern Sicily.
The 2021 administration included all the coastal metropolitan cities of central and southern
Italy (ISTAT, 2020). This is relevant for Civil Protection because Naples, Rome, Palermo
Messina and Catania areas are some of the most densely populated Mediterranean coastal cities
(UNESCO-IOC, 2020; Eurostat, 2022). Adding to these cities also Bari, Reggio Calabria and
Cagliari, also sampled in 2018 and 2021, we reach about 6 million inhabitants.
All these regions were chosen because southern Italy, particularly the Ionian side, have the
highest tsunami hazard (Basili et al., 2021; Basili et al., 2019) compared to other Italian regions
(Liguria, Marche, Abruzzo, Veneto, etc.) with lower hazard. By the way, it is our intention to
complete the survey in the next one-two years in order to have a comprehensive view of the
Italian coastal territory on tsunami risk perception.




|  | 1st Stage | | 2nd Stage | | | 3rd Stage | | | | |
|---|---|---|---|---|---|---|---|---|---|---|
|  | Apulia | Calabria | Molise | Basilicata | Eastern Sicily | Latium | Campania | Sicily (except eastern) | Sardinia | Total |
| Total Residents | 1,716,797 | 1,120,698 | 43,800 | 58,385 | 834,881 | 3,786,704 | 1,925,984 | 2,137,306 | 859,721 | 12,484,236 |
| Coastal Municipalities | 67 | 116 | 4 | 7 | 29 | 20 | 40 | 96 | 71 | 450 |
| No. Respondents | 722 | 491 | 100 | 140 | 374 | 1,034 | 1,170 | 1,221 | 782 | 5,842 |
| Total | 1021 Respondents | | 614 Respondents | | | 4207 Respondents | | | | |
| | 2018 / 2020 / 2021 – Territorial and socio-demographic distribution | | | | | | | | | |
| | Regions | | No. of provinces | | Coastal municipalities | | Respondents | | Km of coast surveyed | |
| | 8 | | 37 | | 69,8% (450/645) | | 5,842 (12,484,236 pop tot) | | 77,9% (6,166km) | |

**Table 1. Sample distribution in the three survey stages**
Table 1 shows the distribution of the sample by survey stage and region. The survey covered a
total of 8 regions, 37 provinces, achieving 69.8% coverage of coastal municipalities for a total
of 6,166 km of coasts surveyed and 5,842 interviews conducted, that are considered
representative of 12,484,236 residents (ISTAT, 2021).
The sampling design was structured respecting robust statistical standards with attention to the
population representativeness of even smaller coastal municipalities. The sample is stratified
by age, gender and coastal regions. In order to have a more statistically robust sample,
education degree was considered as a stratification variable for the third survey stage
(n=4,207).

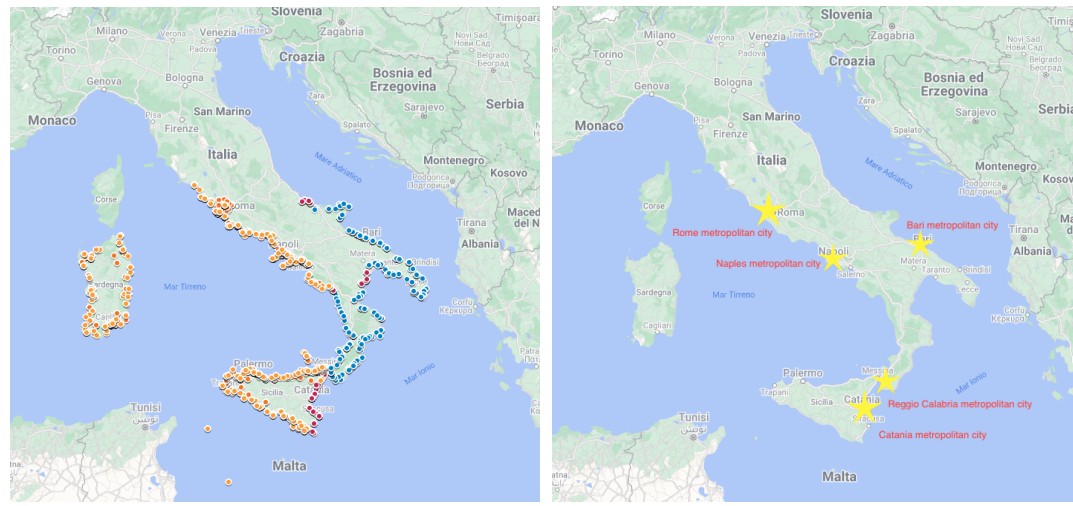

**Figure 1. (Maps Data modified from: © Google Maps, 2022) Maps of the CATI interview distribution (left) and coastal**
**metropolitan cities (right). The different color dots, on the left map, indicate the different survey phases: blue shows**
**the interviews in the first survey phase (2018), red shows the second survey phase (2020), and yellow shows the**
**distribution of interviews in the third survey phase (2021). Yellow stars, on the right map, indicate metropolitan cities**
**(provincial capitals) where population density is high for a wide territorial area.**




**3.3 - Sample validation for the three survey phases (Cronbach's alpha and T-test),**
**statistical data processing**
The three surveys were carried out in different years, this required a greater methodological
accuracy and some preliminary statistical operations to validate and verify the data.
First, we verified that the samples were statistically uniform, independent, and representative
in estimating the reference population mean. Further, whether the datasets could be aggregated
into a single matrix to produce robust outputs and correlations. For this, we used the T-test
(Student, 1908). The T-test (or Students'test) is widely used to study the connection between
natural hazards and risk by considering a specific variable to compare statistical averages
within a group (see e.g., Buylova, 2020; Musacchio, 2021; Liu, 2021). The T-test was applied
to questionnaire's items nr. 8; 12; 14; 16; 20 which are multiple choice questions, and to 21;
23; 24 which are Likert scale questions batteries, between the first and the second surveys
(2018-2020) and in the first and third surveys (2018-2021), respectively. The results of T-test
with a confidence interval of 95% confirm that the samples are statistically uniform, together
correlable and, consequently, analyzable in a single data matrix.
Once obtained the T-test confirmations, we verified that the data collected were consistent and
significant as a part of a robust sample. We calculated Cronbach's alpha (Cronbach, 1988;
1951) on the items comprising questionnaire sections numbers: 2, 3, 4, 5 and 6. The resulting
alpha values between $0,61 \leq \alpha \leq 0,83$ are generally considered optimal to corroborate the
variable reliability (Nunnally, 1975). Particularly, values near $\alpha=0.80$ or greater have an
optimal consistency degree (Peterson, 1994). The alpha values resulting from our comparisons
ranged from $0,74 \leq \alpha \leq 0,89$.
**3.4 - The national sample (National Sample)**
The third survey phase (2021) also included the questionnaire administration to a sample
representative of the whole Italian population distributed by proportional shares.
This has been done through a digital platform that reaches users - who are registered with the
proprietary company - through a link on their smartphones (named TelePanel by the company).
The link allows access to the on-line, re-adapted questionnaire, that users independently
complete. Survey respondents are subscribed to the service and are paid by the commissioning
company. The company, as owner of the sample, takes care that it respects scientific criteria
and that the sample reproduces the same compositions of the population strata. The sample is
generally used to survey shared-interest opinions, political polls, national trends and customs.
Proportional shares are respected and the sample is stratified according to the following
variables: age, gender, geographic area, educational qualification and profession.
The national sample questionnaire surveyed the opinions of 1,500 respondents and was
administered in the same period of the CATI survey.


## 4. Results and discussion

The principal results of the three surveys are presented in two paragraphs: Tsunami risk perception and Knowledge about tsunami.
The map below (fig. 2) shows the geographical positions of the coastal areas where the survey was conducted (by region) and the seas surrounding Italy.

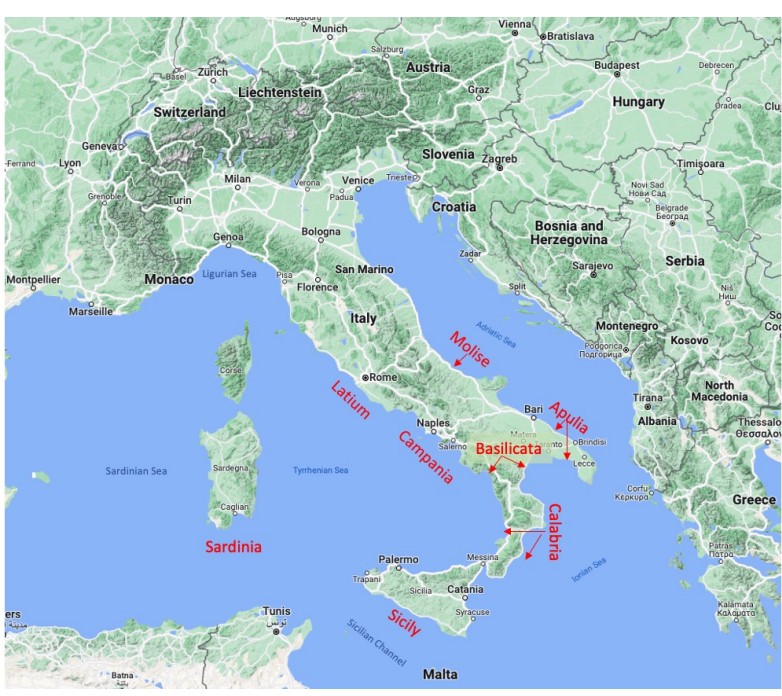

**Figure 2. (Maps Data modified from: © Google Maps, 2022). Map of surveyed Italian coasts and corresponding seas.**

### 4.1 Tsunami risk Perception

The risk perception is preliminary calculated considering the item Q13 "*In the Mediterranean Sea the occurrence of a tsunami is…?*" and Q16 "*Do you think that the coast of your municipality could be hit by a tsunami?*".

Survey data show that tsunami risk perception differs in relation to the seaside.





**Q13 - Possibility of tsunami in the Mediterranean Sea**

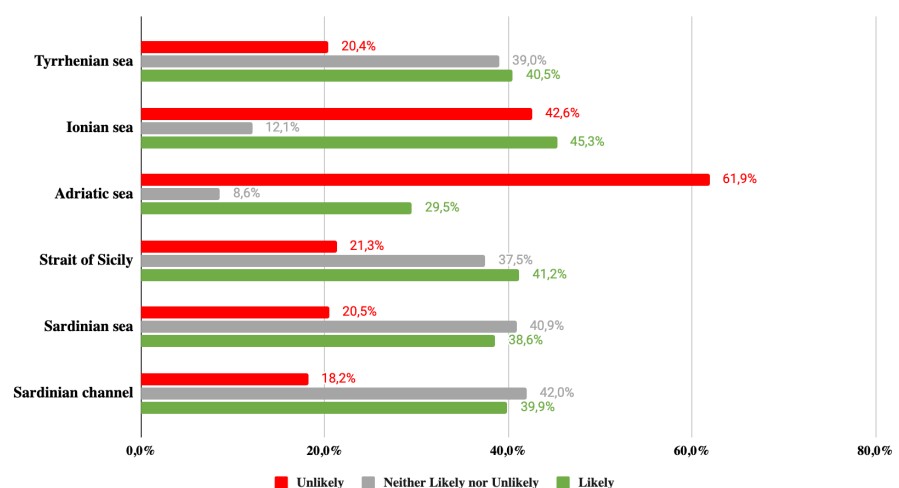

**Figure 3. - Q13 - Tsunami risk perception in the Mediterranean sea by coastal regions**
Figure 3 shows tsunami risk perception for events in the Mediterranean divided by respondents'
coastal seaside. In general, about 40% of the interviewed believe that a tsunami is likely to
occur in the Mediterranean area. While respondents from the Adriatic coast disagree: 62% think
a tsunami cannot occur.
**Q16 - Possibility of your municipality's coasts being hit by a tsunami**

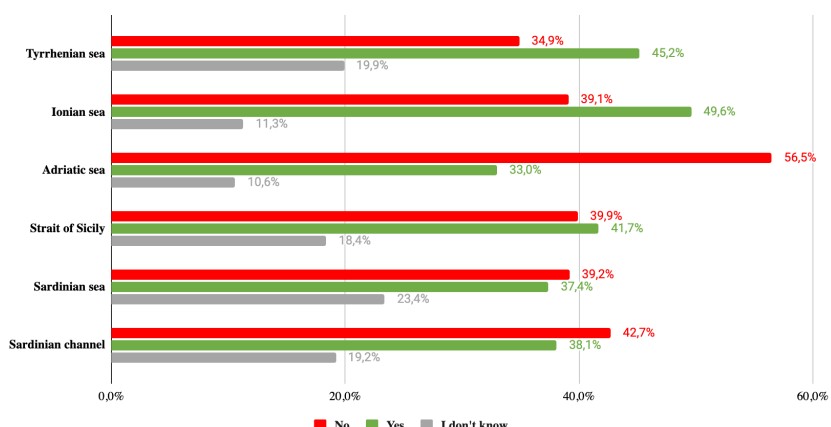

**Figure 4. - Q16 - Tsunami risk perception in respondents' municipalities by coastal regions**



Figure 4 shows the distribution of answers to the specific question about the likelihood of a
tsunami hitting the respondents' municipalities. The sample from the Ionian and Tyrrhenian
sides seem to have a higher tsunami risk perception (49.6% and 45.2%, respectively).
Differently, in the Adriatic coast municipalities, respondents have a lower perception of
tsunami risk, as for the previous question (Fig. 2). In fact, only 33% of them believe that their
municipality may be affected by a tsunami and 56.5% believe it will not.
In the three-stage survey sample (n = 5,842), no significant variations are observed for the
tsunami risk perception in relation to the gender of the respondents.
More significant differences emerge in relation to educational degrees. The sample with low
educational degree showed more uncertainty in responses associated with the "*I don't know*"
modality (23.4% and 23.1%, respectively, versus 11.2% of those with high educational degrees
and 15% of those with medium educational degrees); a high educational degree is correlated
with a significantly higher tsunami risk perception (48.2%) compared to 37.9% for those with
a low educational degree.

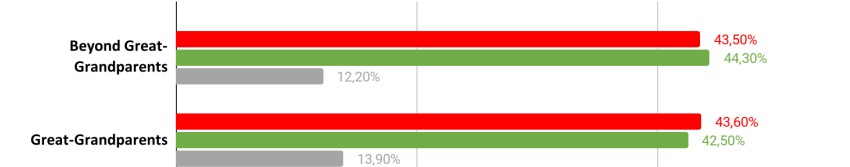

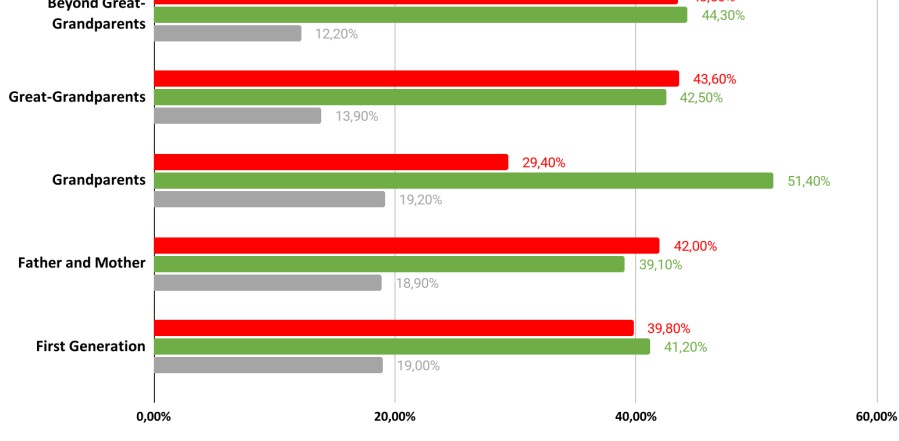

**Figure 5. Tsunami risk perception in municipalities according to the numbers of generations of residence in the area.**
The graph in fig. 5 - reporting the tsunami risk perception in municipalities according to the
numbers of generations of residence in the area - shows an interesting finding. In fact, risk
perception is highest at 51,4% for the third generation (*my grandparents lived there*), whereas
the average response rate for other generations is about 40%. These findings are consistent with
recent studies based on an interdisciplinary historical-anthropological approach (e.g., Garnier
and Lahournat, 2022). Indeed, these studies highlight the role of memory transmission of past
disasters, more generally cultural memory, as an effective tool for DRR (Brokensha et al., 1980;
Fernando, 2003; Gregg et al., 2006; Cohen, 2011; Sutton et al., 2021).



### 4.1.1 Tsunami risk perception in surveyed regions

This subsection moves from the results of our previous paper where a significant difference in
tsunami risk perception between Apulia and Calabria was found, despite the comparable, high
tsunami hazard of the two regions (Basili et al., 2021). In the first region only 30% of
respondents think their region could be hit by a tsunami and 57.5% think it could not, whereas
the results for Calabria are very different, with more than 66% yes and 25% no (Fig. 6).
In order to verify the perception of other regions' inhabitants, we have compared the answers
to the same question (Q16) related to the regions investigated in the second and third survey
phases.
The graph below (fig. 6) shows the percentages of responses collected for Q16 question for all
the regions. The results show a strong heterogeneity among different regions, with the
aforementioned Calabria and Apulia as end-members of risk perception. Molise, adjacent to
Apulia in the Adriatic Sea, is on the low perception side with slightly higher values than this
latter (40% yes, 53% no). Basilicata, with a few municipalities facing on both the Ionian and
the Tyrrhenian coasts, has a slightly higher perception, with 44% yes and 48% no. Moving to
the Central Tyrrhenian Sea, Latium and Sardinia show equal distributions of yes and no with a
large number of "*I don't know*" (40, 40, and 20, respectively). Historical catalogs do not report
relevant tsunamis for these two regions. Southern Tyrrhenian regions (Campania and Sicily)
exhibit higher risk perception (48% and 46% yes, 36% and 33% no, respectively), even if not
as high as Calabria. This can be explained with the presence of known tsunamis (as the 1908
event) or known potential tsunami sources, as active volcanoes of the Neapolitan area and the
Southern Tyrrhenian).
These results are consistent with the similar study, by Gravina et al. (2019). In that case,
inhabitants from southern Italian regions facing the Tyrrhenian Sea were asked: "*Do you
consider to be actively exposed to a tsunami risk?*". 21% of interviewees answered *"highly"*,
more than 42% *"quite"* and over 30% *"low"*. Nonresponses were 9%.
In addition, these regions, including Calabria, are highly seismically active, and people
experienced frequent and even strong earthquakes. The traces of these events, present in the
territory, rises people' s memory from generation to generation.




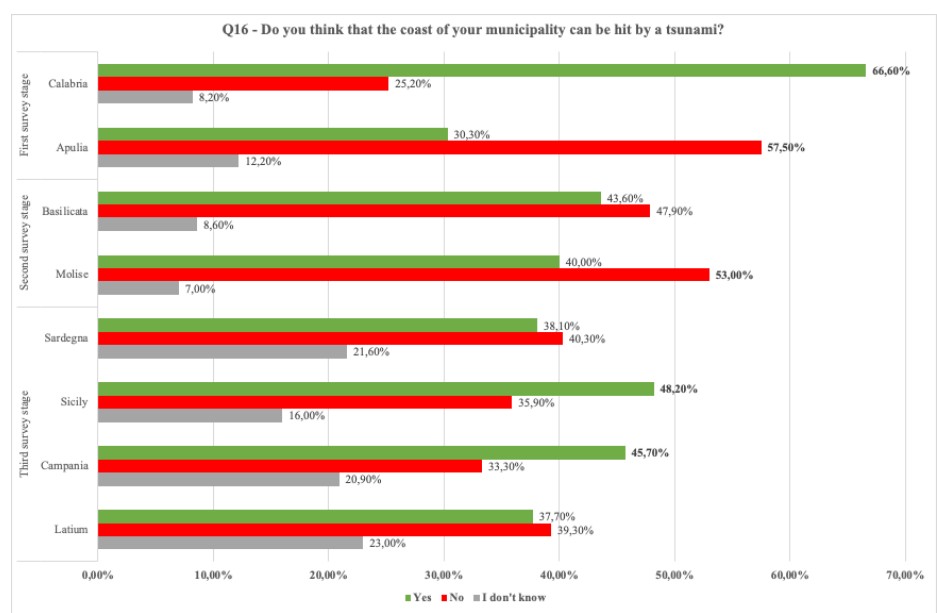

**Figure 6. - Q16 - Tsunami risk perception in respondent's municipalities according to the regions.**

**4.1.2 Tsunami Risk Perception in Metropolitan area and seaside**

We started from the research hypothesis that the perception of tsunami risk could be different
between the inhabitants of metropolitan cities and those residing in the municipalities of the
relative coast. In our opinion, this comparison is particularly relevant for metropolitan cities in
which the exposed value (in terms of human lives, industries and infrastructures) is
considerably higher than the adjacent, less populated coasts.
The following graphs (figures 7-11) show the risk perception surveyed in coastal metropolitan
cities (Rome, Napoli, Bari, Reggio Calabria and Catania) in relation to the risk perception of
the seaside on which the city lies (tab. 2).

| Region | Latium | Campania | Apulia | Calabria | Sicily |
|---|---|---|---|---|---|
| **Metropolitan coastal city** | **Rome** | **Naples** | **Bari** | **Reggio Calabria** | **Catania** |
| **Sub demographic areas** | 6 | 5 | 2 | 5 | 2 |
| **Total municipalities** | 121 | 92 | 41 | 97 | 58 |
| **Total residents** | 4,342,000 | 4,250,000 | 1,261,000 | 549,000 | 1,068,000 |
| **Resp by metropolitan city** | **824** | **938** | **169** | **134** | **155** |
| **Resp by seaside** | **3201** | **3201** | **549** | **910(Ioni)/3201(Thyrr)** | **910** |

**Table 2. Demographic data of in-depth metropolitan areas and sample size by city and seaside (Source ISTAT, 2022.**
**Database accessed on 07/2022 at: http://dati.istat.it/index.aspx?lang=en&SubSessionId=2c2456e1-fbc4-45fa-bdd3-**
**bab2caf314ac)**

To prevent statistical bias the data for each individual metropolitan city were removed from
the coastal data on which the city is located. We also carried out the T-test for independent
samples from which a significant difference between the averages of the two samples



(metropolitan cities and coastal slopes) were found. We used the test to indicate the sample
statistical uniformity and comparability.
The graph in Figure 5 shows the comparison between the sample of the Metropolitan city of
Rome and the relative Tyrrhenian coastal slope. In the case of the metropolitan city of Rome it
is important to remember that Rome (2.7 million inhabitants) cannot be considered entirely a
coastal city, even if some densely populated districts (like Ostia, with its 231,000 inhabitants)
are entirely located on the seaside.
This is probably the reason why the average of the tsunami risk perception in Rome (fig. 7)
seems to be lower than the Tyrrhenian. Indeed, only 36.3% of the respondents believe that a
tsunami may hit their municipality, versus the 48,2% of respondents living in the Tyrrhenian
seaside.

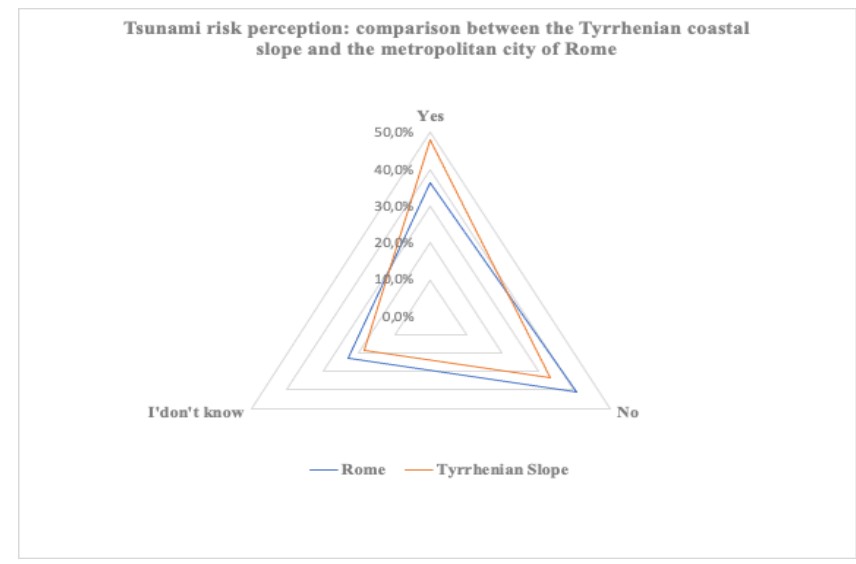

**Figure 7. Tsunami risk perception: comparison between the Tyrrhenian coastal slope and the metropolitan city of Rome. (Q-16. Do you think that the coast of your municipality could be hit by a tsunami?)**

Tsunami risk perception in the Naples metropolitan city (fig. 8) is slightly higher than the
average surveyed on the Tyrrhenian side. 47.2 % of respondents said that a tsunami could hit
the coasts of their municipality compared to the Tyrrhenian slope average, which - excluding
the metropolitan city of Naples - has a value of 44.9 %.
31.8% of respondents believe that a tsunami is unlikely to hit their municipality's coast, and
21% say they don't know the answer.
These data are consistent with the seaside average and it diverges from the starting hypothesis.
These data could be related to multi-hazard variables that lead the municipalities' residents to
express a greater risk perception due to the presence of the Vesuvius volcano, the seismic
memory of the 1980 Irpinia earthquake, and the frequent bradiseisms and micro-earthquakes
that occur in the Neapolitan area (i.e., in the Phlegrean Fields). Traces of these events are also
found in the literature, such as the case of the tsunami of 1345, documented by the Italian
famous writer Francesco Petrarca in his "*Letters on Familiar Matters*" (Bernardo and Petrarca,
1985) , and recently described in Rosi et al. (2019).
Moreover, the city of Naples and the coastal municipalities of its hinterland, have high exposure
due to civil settlements and industrial complexes including some Major Accident Hazard
Industries (Tinti and Armigliato, 2003; De Pippo et al., 2008; Grezio et al., 2012) that are
located on the coast. Last but not least, the metropolitan city of Naples has one of the highest
coastal population densities in Europe, with concentrations ranging between 500 and 2,500
inhabitants per $km^2$(ISTAT, 2020).

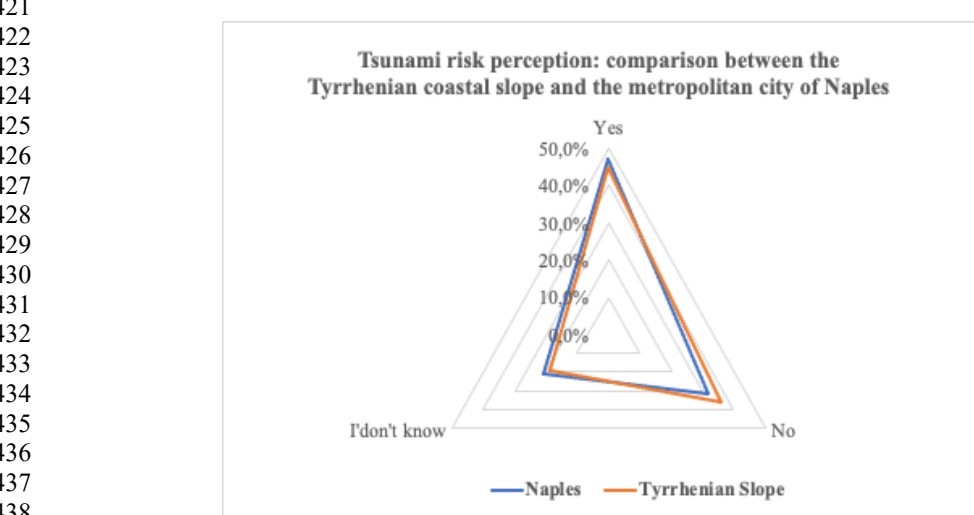

**Figure 8. Tsunami risk perception(Q-16): comparison between the Tyrrhenian coastal slope and the metropolitan city**
**of Naples.**

The average risk perception for the metropolitan city of Bari is low (fig. 9), in line with the
Adriatic coast. In fact, we observe that only 34.3 % of respondents believe that a tsunami could
hit their municipality, versus 32.4 % of those residing in municipalities on the Adriatic coast.
We also observe that 54.4% of respondents say a tsunami could not hit their coastal
municipality, compared to 57.4% of the average of those living in coastal municipalities on the
same seaside.
The perception results appear to be low compared with the estimated hazard for the southern
Adriatic coast and Bari metropolitan city, which is medium/high (Basili, et al., 2021). The
estimated hazard takes into account the strong earthquakes occurring along the Hellenic arc,
able to generate tsunamis that would hit the Adriatic and Bari coasts. The low tsunami risk
perception may also be influenced by the absence of recent tsunami events (Maramai et al.,
2019) as already noted in our previous paper.


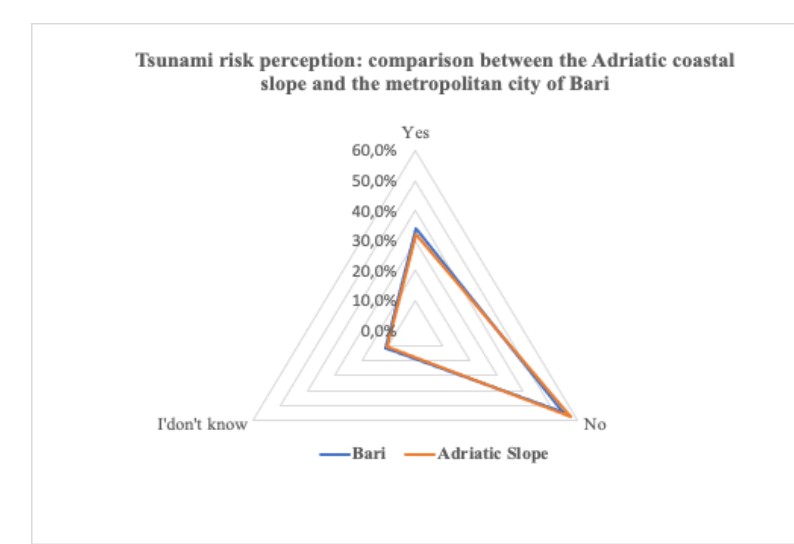

**Figure 9. Tsunami risk perception (Q-16): comparison between the Tyrrhenian coastal slope and the metropolitan city of Bari**

The tsunami risk perception in the metropolitan city of Reggio Calabria is on average high (fig. 10). The graph shows that 70.9% of respondents believe that the city may be hit by a tsunami compared to the average of 46.5% of respondents living in the Ionian slope's remaining municipalities and 45% of those living in the Tyrrhenian slope's municipalities. Furthermore, data analysis shows that only 8.9% of respondents answered "*don't know*" to the specific question. This percentage could indicate that residents of the metropolitan city of Reggio Calabria, have a greater tsunami risk knowledge of their area, compared to the other metropolitan cities considered in the analysis. The high tsunami risk perception is likely related to the 1908 tsunami, which had a strong impact on the territory, causing widespread damage and about 2,000 casualties (attributed to the tsunami), which still today holds a high media echo, currently in resident's memory.

In addition, residents' tsunami risk perception is in line with the high tsunami hazard estimated for the Reggio Calabria area (Basili et al., 2021).




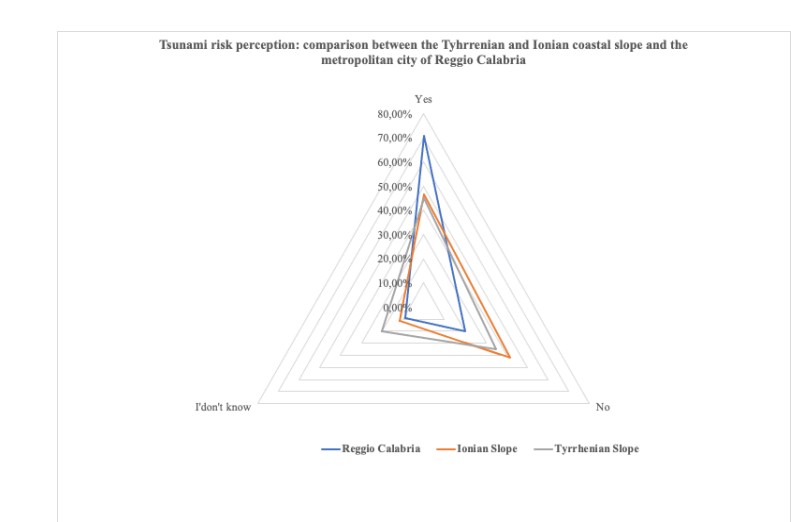


**Figure 10. Tsunami risk perception(Q-16): comparison between the Tyrrhenian coastal slope and the metropolitan city of Reggio Calabria**


The tsunami risk perception in Catania metropolitan city (fig. 11) is slightly higher (49,7%)
than the risk perception of respondents of the Ionian coastal slope (45,8%). Only 11,6% of the
respondents answered *"I don't know"*, in line with the responses from the other coastal slope
municipalities. The percentage of tsunami risk perception in Catania, is probably associated
with the presence of easily recognized hazards (e.g., volcanic risk and ash management due to
the close Etna volcano, frequent earthquake shaking, etc.). In addition, industrial complexes
and refineries along the coast increase the exposed value, and possibly the risk perception Not
least, the tsunami hazard (Basili et al., 2021) in the Catania area is quite high, due to both local
and distant tsunamis.

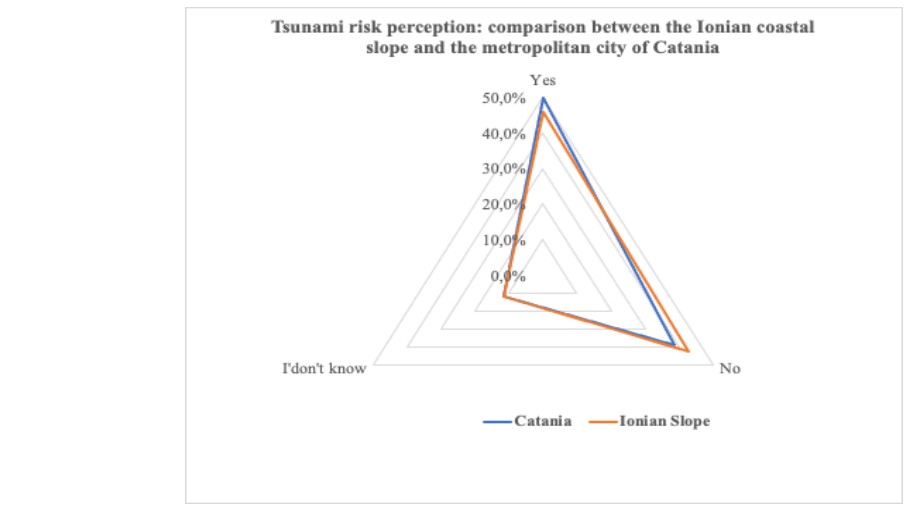


**Figure 11. Tsunami risk perception (Q-16): comparison between the Ionian coastal slope and the metropolitan city of Catania**



Comparison among metropolitan cities (fig. 12) shows a higher tsunami risk perception in
Reggio Calabria, Catania, and Naples. These cities, throughout history, have been repeatedly
affected by disruptive natural events including strong earthquakes, volcanic eruptions and
tsunamis. The difference in risk perception between metropolitan areas and coastal slopes
remains the subject of further study. Bari metropolitan city has a low tsunami risk perception
even though it is located in a stretch of coastline where tsunami hazard is considered
medium/high. This could be associated with events that occurred in the distant past and the low
frequency of occurrence of earthquakes.

**Q-16. A comparison between tsunami risk perception in metropolitan cities**

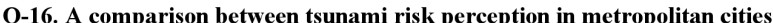

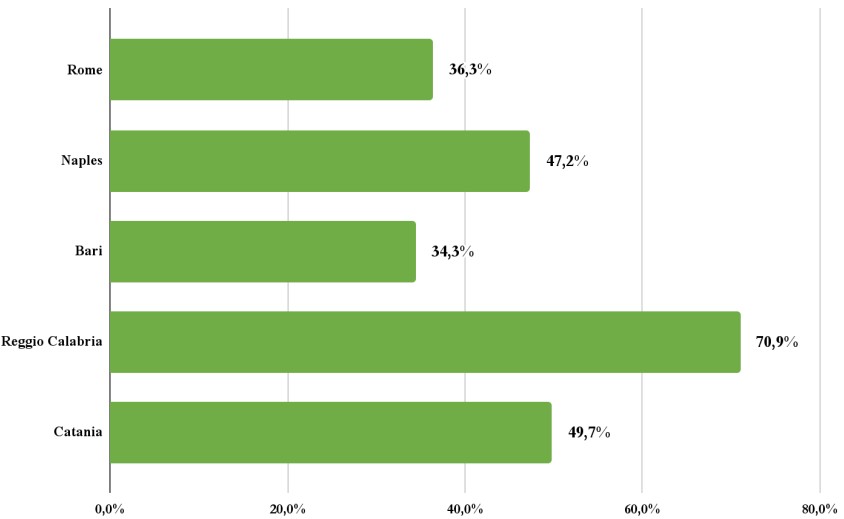

**Figure 12. Tsunami risk perception (Q-16): comparison between the metropolitan cities**
**4.1.3 Tsunami risk perception: comparison with a national sample**
In this section, we compare the tsunami risk perception in the Mediterranean area (Q-13),
surveyed among the whole coastal population and a national population sample (National
Sample survey n = 1,500). From the graph (Fig. 13), it can be seen that risk perception is higher
in coastal communities (39.4%, CATI survey) and lower in the national average (19.2%
National Sample survey). Minor variations can be observed in the other response modes. A
difference of 13% also emerges in the central mode "*Neither likely nor unlikely*" from which
low tsunami knowledge is assumed.
The National Sample survey becomes of primary relevance to also investigate the risk
perception of the population not living in coastal areas who might face this risk in a summer
vacation context even in non-national territories. This result is not surprising, considering the
lower familiarity of non-coastal inhabitants with sea activities - and hazards - compared to
coastal cities residents. Also, we should consider that the tsunami risk for people spending for
instance a two-week's vacation in a seaside location is statistically much lower compared to



the risk to which a coastal resident is exposed. We anticipate that this result strongly suggests
the need for a communication effort specifically oriented to tourists.
These data can be considered representative of the national mean related to the tsunami risk
perception and may be used for comparison with data related to the same specific groups of
population living on the coasts.

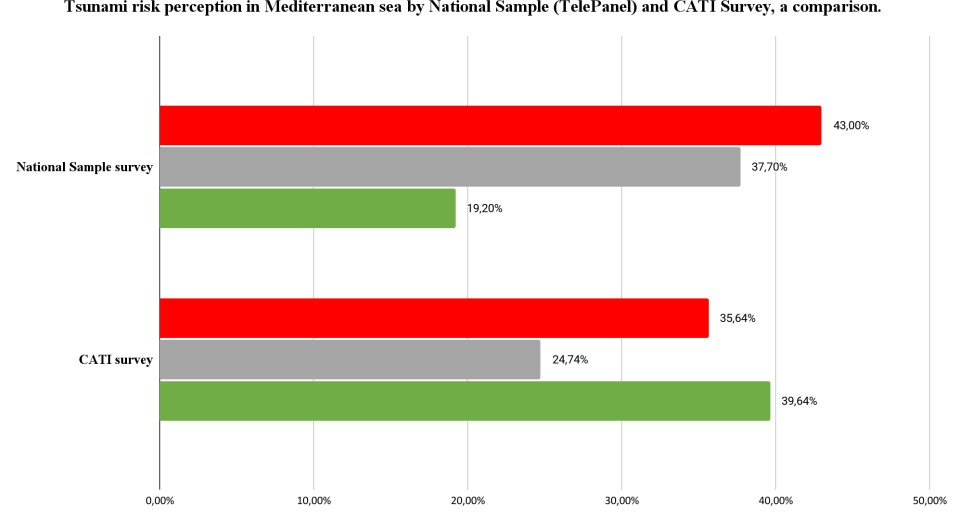

**Figure 13: Tsunami risk perception in Mediterranean Sea by National Sample and CATI Survey, a comparison.**
**4.2 Tsunami Knowledge**
**4.2.1 Phenomenon description: elicitations of the terms tsunami and maremoto**
In order to deepen understanding of people's tsunami knowledge we continued the
investigation on qualitative attributes of the tsunami phenomena started the our previous paper.
The first step is aimed at exploring the differences between the Japanese word "*tsunami*",
broadly used in the international scientific community, and the word "*maremoto*" (literally
seaquake), being a common alternative in colloquial Italian language. The overall results of
this research confirm the different meanings attributed to these two terms by respondents.
Figure 14 shows that the largest part of the sample shows greater familiarity with the term
"tsunami" (57%) while "maremoto" drops by several percentage points (43%). The word
"tsunami" seems to sound more familiar for those who have higher education levels (66%) and
less familiar to people over 65. Some interesting differences regarding the usefulness of the
two terms are related to local characteristics and will be further investigated. For example in
Reggio Calabria metropolitan city interviewees generally use the term maremoto to identify
the phenomenon (57%) whereas in Naples the term tsunami is more familiar (63%).
We could hypothesize that past events do differently shape the way the phenomenon is
acknowledged and understood, as culture provides different resources to address these events,
where traditional environmental knowledge plays a prominent role along with scientific





communication. These aspects are reflected in social representations (Moscovici, 1961) as well
as in language being used to express such representations (Moscovici, 1976).

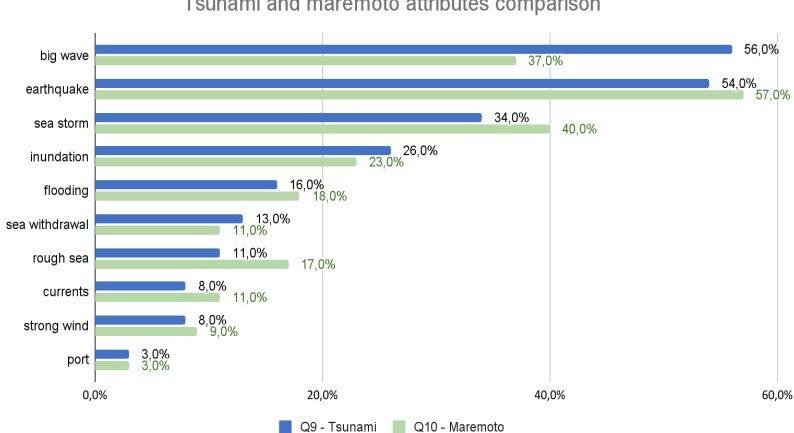

**Figure 14: Tsunami and maremoto attributes comparison**
Data in fig. 12 show that the word "tsunami" is mostly associated with *big wave* (56%),
*earthquake* (54%) and *sea storm* (34%), and in a more detached position *inundation* (26%),
*flooding* (16%) and *sea withdrawal* (13%). Instead, the word "maremoto" is first associated
with *earthquake* (57%), then with *sea storms* (40%) and *big waves* (37%). *Inundation* (23%)
and *flooding* (18%) are still present along with *rough sea* (17%).





### 4.2.2 - Knowledge about causes of tsunamis

With regard to the alleged causes of tsunamis, they are generally attributed correctly to earthquakes 74% as the main cause, then to volcanic eruptions (44%), t in agreement with the most frequent tsunami causes worldwide (NCEI/WDS, 2022) and in the Mediterranean (Maramai et al., 2014).

In general, it is observed that tsunami knowledge is not directly related to the gender and distribution of the interviewees but directly increases with instruction level and decreases with age (young, highly educated people under 50 years old are best informed).

Meteorological phenomena t are mentioned by 19% of the respondents. In general terms this is is somehow overestimated, but we know that meteotsunamis are rather frequent in the Mediterranean, especially in the Adriatic Sea (Šepić et al., 2009; Vilibić et al., 2009; Maramai et al., 2022). However, it is possible that people confuse sea storms with (meteo)tsunamis. To verify this, we carried out a bivariate analysis that can provide some clues for this belief. Indeed, data highlight higher percentages for those who live in municipalities overlooking the Sea of Sardinia as well as the Channel of Sardinia which are frequently swept by strong winds of Maestrale, which can locally originate sea storms with waves several meters high.

We also note that weather conditions are most frequently mentioned by people with low education levels (22%) versus 15% of the sample with high educational level.

Furthermore, landslides are properly indicated as a possible tsunami caused by 14,8% of respondents, as well as meteorites and space objects. A similar question, about tsunami risk perception induced by rock landslides, was asked in the survey conducted in Norway during the ASTARTE project by Goeldner-Gianella (2017). In Norway, respondents correctly show high tsunami risk perception induced by rock landslides. This result is mainly due to the frequency of rock collapses occurring locally, and to citizens' trust in local institutions for how tsunami risk is managed and how information is disseminated (Goldner Gianella et al., 2017).

### 4.2.3 - Knowledge about tsunami effects

Knowledge about tsunami effects are investigated by Q22 (*Try to figure out the effects of a tsunami / tsunami on the coasts of your region. How far do you agree with the following statements?*) Results show that in general, people are well aware about tsunami possible effects on the coasts of their region. In particular, deaths and serious injuries are recalled by 93,9% of their sample; damages to houses, buildings and infrastructures by 93.1%, and negative impacts on economy and occupation and on economy are both indicated by 89,4%. Worth to say that despite high rates of knowledge about tsunami possible impacts, women, most educated people, and people aged up to 65 years show slightly higher percentages, as well as the inhabitants of the municipalities included in the area hit by the 1908 Messina Strait tsunami. As to resume, these kinds of effects are well known and present to interviewees.

The data also show a catastrophic representation of tsunami effects that does not correspond to the expected effects on the Italian coasts where moderate-sized tsunamis are more likely to occur. This scenario probably comes from the diffusion of the catastrophic images of the tsunamis that occurred in Sumatra in 2004 and in Japan in 2011, widely conveyed by television and social media. Given that macro-effects of a tsunami are well recognized and understood, micro-effects at local and individual level seem to appear unfamiliar to respondents.

Indeed data show that large coastal flooding is acknowledged by 83% and sea withdrawal by 77%, and the possibility that a small tsunami might drag an adult into the sea is known by 75% (Q23). Only 24% of respondents think that fleeing to the beach after a strong earthquake is an





appropriate behavior. The question about the possibility that a large tsunami with waves up to
20 meters may occur in the Mediterranean Sea raises some concerns, as only 46% of the overall
sample considers it as a real possibility. Cross-tables highlight some unexpected surprises, such
as higher rates of youngsters (31%) and women (27%) who consider fleeing to the beach a
proper response to a massive earthquake.
**4.2.4 - Data on the sources information**
Data on the sources of information being used by the interviewees provide a relevant
framework to address and improve tsunami risk communication. As for the previous paper, we
have decided to group different sources into homogeneous categories. Data provide a clear
indication on the central role of television, that is indicated as an information source by almost
90% of the sample. More in detail, TV news reaches 83% of respondents and documentaries
or scientific channels reach 23%. Other traditional broadcast media, considered as a unique
category including newspapers, books, radio and movies, are found to reach 58% of the public.
Considering disaggregate penetration rates, newspapers were mentioned by 35%, books by
19%, movies by 12% and finally radio had 8%. Surprisingly enough, the whole internet sources
show a penetration rate of about 18% and interpersonal sources such as friends, relatives and
neighbors weigh for 5%.
As noticed in the first step of the research, the impact of institutional and scientific sources
appear to be a problematic issue urging the development of a proper and effective risk
communication strategy, since their overall penetration rate is a mere 3,5%. More in detail,
single rates are about 2% for the Civil Protection, 1,6% for Universities and Research
Institutions and barely a 1% for local administrative entities such as the Region, the Province
and the Municipalities. The residual category other has been mentioned by only 35 people,
corresponding to 0,6%. This suggests that a strong effort is needed for institutional parties to
fill this gap, offering more capillary information using state-of-the-art communication
channels.
Of course, data deserve further analysis by means of data reduction procedures, aimed at
aggregating variables into new indicators and producing synthetic more effective
understanding of the considered phenomena.



## 5. Discussion and conclusive remarks

The three surveys on tsunami risk perception, conducted between 2018 and 2021, started from the necessity to study and understand the level of knowledge of tsunami risk and the awareness of Italian citizens living in or visiting coastal areas exposed to tsunami hazard.

To date, these surveys represent a relevant sample of the Italian population, both for the number of interviewees and for the adopted methodology (5,842 CATI interviews carried out on over 6000 km of coastline with moderate to very high tsunami hazard, plus a nation-wide "telepanel" sample representative of the whole Italian population).

The main results show that the tsunami risk perception varies significantly according to the coastal region. In particular, regions in the Adriatic (Apulia, Molise) show very low levels of risk perception compared to Calabria, Sicily and Campania, both on the Ionian and the Tyrrhenian seaside. Latium and Sardinia lie in the middle, with equal numbers of people thinking that a tsunami could hit their region.

Educational degree affects tsunami risk perception indeed: the higher the educational degree, the higher the tsunami risk perception.

On the contrary, data analyses show that tsunami risk perception is not influenced by either gender or age. A slight difference is observed in the middle age group in which interviews of 35-49 years appear to have a slightly higher perception. The elderly show the lowest frequency percentage in the response modality *"I don't know"* associated with low tsunami risk perception.

These data agree with several studies carried out on local cultures have shown how communities that have experienced disastrous natural events in their past, developed better resilience and preparedness (Dekens, 2007) to these calamitous events. Such knowledge, which differs from scientific knowledge (Flavier et al., 1995), is associated with the historical memory of past experiences learned and transmitted through rituals, traditions, narratives and folk songs (e.g. Smong Song in Simeulue Island) (McAdoo et al., 2006; Rahman et al., 2017; Rahman et al., 2018; Sutton et al., 2021) and defined in a different way as: "local knowledge", "traditional knowledge", *"indigenous technical knowledge", "peasants knowledge", "traditional environmental knowledge" and "folk knowledge"* (Sillitoe, 1998; Mercer et al., 2007; Mercer et al., 2012).

Memories of previous disasters both inform people's knowledge of their environment and vulnerability and also influence their risk interpretation and response to future disasters (Arunotai, 2008). Collective memory, relying on oral tradition shared by a specific group, most commonly the family, tends to disappear with the death of the last eyewitness to the event (the three-generation limit). Cultural memory, supported by documents (such as newspapers, archives, images) and memorials as tangible signs for the community, ensures that disaster meanings and interpretations are recorded and transmitted from generation to generation. These forms of cultural memory, considered by Mercer (Mercer et al., 2010) as an existing or acquired knowledge set by local communities are born and maintained through the accumulation of experiences, social relations, community practices and institutions, and their transmission across generations.

Garnier and Lahournat's (2022) study highlights how Japan Stone Monuments, representing elements of both tangible and intangible culture for the population, demonstrate the existence of disaster memory and reflect a desire to commemorate and transmit significant past events to current and future generations. These findings highlight the importance of oral transmission between generations about tsunami risk and could be very useful for designing effective

information and communications activities about tsunami risk reduction (Spahn et al., 2010;
Løvholt et al., 2014; Oktari et al., 2018; Sutton et al., 2020).
In the work we also carried out a comparative analysis on tsunami risk perception between the
metropolitan areas and the respective coasts (see paragraph 4.1.1), starting from the hypothesis
that risk perception could be different among the population residing in a large city and the
population of small municipalities distributed along the seaside. The territorial units analyzed
were selected based on exposure (higher coastal population density or proximity of major urban
settlements to the coast) and territorial vulnerability (high concentration of anthropogenic
activities such as industries or intensive tourism activities) according to ISTAT (Italian national
statistical institute) data.
The goal of this comparison is both to highlight possible differences in perceptions associated
with densely populated urban areas, and to provide a solid basis for targeted risk mitigation
actions in specific contexts (for example to improve and better address the Tsunami Ready
Program - Valbonesi et al., 2019; Valbonesi, 2022).
The results indicate that in Reggio Calabria, Catania and Naples indeed there are significant
differences in the tsunami risk perception compared to the relative seaside. This data is most
evident in Reggio Calabria (71%) with a difference of over 25 percentage points compared to
the Tyrrhenian (45%) and Ionian (46%) sides of the same region (Calabria).
The data continues to be significant, albeit less evident, for Catania (50%) and the Ionian
seaside (46%) and for the city of Naples (47%) and the Tyrrhenian seaside (45%). It is useful
to underline that the metropolitan city of Bari, despite being in a stretch of coast considered to
be at high risk, has significantly lower values of perception of the tsunami risk (34%), in line
with the data for the whole Adriatic coast (32%). Tsunami risk perception in the metropolitan
area of Rome seems to be lower than on the Tyrrhenian coast. Probably this is because Rome
cannot really be considered a coastal town because most of its over 3 million inhabitants live
far from the sea. The presence within the metropolitan area of Rome of the town of Ostia, a
very populous municipality (over 231,000 inhabitants) lying along the Tyrrhenian coast, is not
enough to raise the overall risk perception of the capital. It would be interesting to deepen this
aspect with specific analysis of the local residents, commuters and visitors. Given the city's
proximity to the coast (about 20 km), it is also worth considering the large daily (as well as
seasonal) tourist flows that in the summer period occur.
The Telepanel data (n = 1,500), collected in the same time period as the third survey, deserve
particular attention because they are a representative sample of the population at the national
level, that means people predominantly living far from the seaside. Comparing the tsunami risk
perception of the Telepanel with the surveys carried out on the coastal population, it emerges
that the coastal population on average has a perception of risk significantly higher than the
national average of about 20 percentage points. These data suggest that it is necessary to pay
particular attention to the coastal tourist areas where every year millions of people spend their
holidays. Providing capillary information along the beaches and in the harbors, indicating
escape routes and meeting points, establishing redundant alarm systems, are the main tools to
be put in place to reduce the tsunami risk of both residents and tourists.
As for the knowledge of tsunamis, we started by considering the different associations linked
to the two terms that are commonly used in the Italian language: tsunami and maremoto.
Results indicate that most of the sample is more familiar with the term tsunami (57%). The
word tsunami is more familiar to those with a higher level of education (66%) and less familiar
to people over 65 years old.


However, there are some differences that appear to be linked to local characteristics and which
will be the subject of further investigation.
It is interesting to notice that the two terms are associated with different phenomena. For
instance, the term tsunami is mostly related to the occurrence of a great destructive wave, while
the word tsunami is more often related to the occurrence of an earthquake.
As for the knowledge of the causes that generate tsunamis, in general the sample correctly
attributes the occurrence of a tsunami to earthquakes and to volcanic eruptions, in line with
scientific knowledge. Furthermore, tsunami knowledge does not seem to be linked to the
gender and the areas of residence of the interviewees but increases according to the educational
degree and inversely to age (people with higher education degree and younger than 50 years
old appear to be more informed).
Our data indicate that in general people are aware of the possible effects of tsunamis on their
regions. However, this knowledge appears to be closely influenced by the media representation
of the great tsunamis that occurred in Japan (2011) and Sumatra (2004). This type of media
representation could turn out to be misleading with respect to the more modest phenomena that
can be generated by smaller tsunamis, more frequently expected in the Italian coastal territory,
but also capable of causing serious damage and victims.
Moreover, our data indicate that television continues to play a central role in conveying
information relating to the tsunami risk, while the information role played by social media and
the web appears to be still marginal. A particularly problematic aspect concerns the poor
visibility of scientific institutions, civil protection and local authorities to be recognized by the
interviewees as official sources of information on the tsunami risk.
Our results suggest that the loss of memory of past events affects the perception of risk by
citizens and communities, making more difficult and even ineffective risk mitigation actions
(see for example Kurita et al, 2007; Sugimoto et al. 2010; Arias et al., 2017 and Wei et al.,
2017). Memory and recollections (such as commemorations of past events) are relevant for the
development of risk mitigation strategies and to increase the population resilience. More
specifically, a work of awareness raising aimed at attributing a sense and a meaning to the
memory is needed in order to reduce the risk. We believe that the results of this study, although
limited to central and southern Italy, can be used in other countries of the NEAM region and
worldwide to orient communication strategies and risk reduction actions.
In the near future, given the large amount of data collected in the three surveys, we will focus
on the creation of synthetic indexes for the perception and knowledge of tsunami risk.
We are also working to deepen the knowledge related to the cultural differences in the
perception of risk which seem to be very influenced by the local culture of reference. We are
confident that this will allow us to better explain the differences in perception and knowledge
that the data show in the different slopes and metropolitan cities. Finally, our efforts will be
focused on translating the results of risk perception analyzes into effective communication
strategies for tsunami risk reduction. Not least, the 2018 pilot survey results - published in in
the our previous paper - were extensively used in developing and improving the CAT website
content. Moreover, several relevant aspects of the study made it possible to better address
approaches in risk awareness campaigns such as "Io Non Rischio" (a Civil Protection
campaign) as well as undertake dissemination campaigns aimed both to raise awareness and
survey tsunami risk perception in schools (one of the Tsunami Ready program indicators).



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
