# Peer review of "Tsunami risk perception in Central and Southern Italy."

_Natural Hazards and Earth System Sciences, 2022_

## Referee Comment (RC2)

[referee-annotated manuscript omitted]

---

## Author Response (AR1)

**Tsunami risk perception in Central and Southern Italy.**

**Lorenzo Cugliari[1], Massimo Crescimbene[1], Federica La Longa[1], Andrea Cerase [1,2], Alessandro Amato[1], Loredana Cerbara[3]**

[1]**National Institute of Geophysics and Volcanology, 00153, Roma, Italy**
[2]**Department of Communication and Social Research, La Sapienza University, 00198, Roma Italy**
[3]**Institute for Research on Population and Social Policies, National Research Council, 00185, Roma, Italy**

*Correspondence to*: Lorenzo Cugliari (lorenzo.cugliari@ingv.it)

**DOI: https://doi.org/10.5194/nhess-2022-212**

**Answers to reviewer's questions 1**

| Question |
| --- |
| 1- I cannot understand "capillary information"(Line 694, 789). If you intend "evacuation information", that makes sense. |
| **Answer** |
| Thank you for the note. We rewrite the sentence in this way:
 This suggests that a strong effort is needed for institutional parties to fill this gap, through widespread and comprehensive dissemination of information (including info on evacuation routes and procedures)  by using state-of-the-art communication channels. |
| **Question** |
| 2- UNESCO-IOC-ITIC have recommended such tsunami education technical words "evacuation" routes, not "escape" routes (Line 789). It is better for you to follow the tsunami technical words in this paper (see following URL: http://itic.ioc-unesco.org/index.php?option=com_content&view=category&id=1406&Itemid=2280). |
| **Answer** |
| Thank you for your suggestion. As indicated in UNESCO documents, we replaced "escape route" with "evacuation route". |

**Answers to reviewer's questions 2**

| Q |
|---|
| **1-** How far inland do you assume a coastal belt? This could also give an idea of the population density. In the Rome case, the importance of this distance looks important. |
| **A** |
| Thanks for the note.
To define coastal areas we used the parameters given by ISTAT (Italian national statistical institute). They provided the following definition:
The coastline is defined as the line where the land surface meets the sea. The average of the high tide is used to delineate the EU coastline. The coastline is the territory that is at most 10 km from the coastline (ISTAT, 2020; DOI: 10.1481/Istat.Rapportoterritorio.2020). PeRegarding Rome, preference was given to the population residing in coastal municipalities. In addition, the 10th district of Rome (Ostia Lido) has more than 230,000 inhabitants. |
| **Q** |
| **2-** Have you considered interviewing people inland to know their perspective, but also since they could be, most probably, possible coastal visitors? |
| **A** |
| Yes, in the future that would be interesting. For now, we surveyed a population sample living on the coasts of 8 regions and also surveyed 1,500 interviews with a sample (Telepanel with CAWI methodology) distributed over the whole country. The results will be published in future articles. |
| **Q** |
| **3-** A clear distinction of hazard and risk should be made. I think in lines 113-114 there is some confusion. |
| **A** |
| Generally, risk perception studies don't take into account the factors used to define risk (Hazard, Exposure, Vulnerability). In our work we try to do this to relate physical dimensions of the event with risk perception, as to better integrate risk perception studies within a multidisciplinary research context. Risk perception includes all the questionnaire items regarding hazard, exposure and vulnerability of the Italian coast that could be affected by the tsunami.
In the text, at lines 113 and 114 we have tried to clarify this concept. |
| **Q** |
| **4-** How did you unsure and homogenize the results of three different surveys in time and the national survey as well, which I understood that is was a separate survey form CATI, only to |

the very end (lines 706-708), where it makes clear that the national survey is not included In the CATI sample. The use of the national sample is not clear enough (section 4.1.3).

**A**

In Section 3.3 we described the statistical methods used to validate and unify the three CATI survey phases. Regarding the National sample (section 3.4), it is built under different criteria due to different nature of interviewees selection and it is used only as a term of comparison (see section 4.1.3).To explain this passage we have modified line 232 and inserted 245-246 and lines.

**Q**

**5-** Socio-demographic criteria were also applied to all the surveys or only to the national survey. Please clarify somewhere in the text. I understood that age was applied, but I am not sure about education level for example.

**A**

Thanks for the suggestion:
Yes, we always considered all socio-demographic variables for data analysis (Gender, geographical areas and educational degree) as mentioned in lines 197-201. Education level quotas was considered only for the third phase of the survey (CATI survey) and for the national sample (Telepanel). The reason is that in the first and second phases the amount of the samples were non enough for the application of education quotas, but this was not problematic from a reliability point of view and in the unification of the three surveys.

**Q**

**6-** In the paper Q16 is extensively discussed. The reasons for the outcomes of this question were only speculated and assumed (for example: "The percentage of tsunami risk perception in Catania, is probably associated with the presence of easily recognized hazards" or "The high tsunami risk perception is likely related to the 1908 tsunami"). Were there any other questions included in the survey, which could help identifying the levels of risk perception that were reported based on Q16?

**A**

No, the questionnaire does not contain specific questions aimed at understanding the causes of increasing or decreasing risk perception in the surveyed areas. The hypotheses are based on the diverse expertise of the research team, which also conducts qualitative, historical, and cultural studies on tsunami risk knowledge and perception. The questionnaire also has a section on cultural theory and we are analyzing results that will be the subject of another publication.

**Q**

**7-** "the tsunami hazard (Basili et al., 2021)" is mentioned for every region analyzed in section 4.1.2, but in a very brief (one sentence) way. A few more words would be very useful to link the risk to the hazard.

**A**

We have not included more details on the comparison between hazard and risk perception because we are working on creating synthetic indicators that can resume and aggregate different questions, to facilitate and refine the comparison between different risk factors such as Hazard, Exposure, Vulnerability